# Highly Efficient All-Solution-Processed Quantum Dot Light-Emitting Diodes Using MoO_x_ Nanoparticle Hole Injection Layer

**DOI:** 10.3390/nano13162324

**Published:** 2023-08-12

**Authors:** Ji-Hun Yang, Gyeong-Pil Jang, Su-Young Kim, Young-Bin Chae, Kyoung-Ho Lee, Dae-Gyu Moon, Chang-Kyo Kim

**Affiliations:** Department of Electronic Materials, Devices and Equipment Engineering, Soonchunhyang University, Asan 31538, Chungnam, Republic of Korea; wlgns123789@sch.ac.kr (J.-H.Y.); schqled@sch.ac.kr (G.-P.J.); suwim8549@sch.ac.kr (S.-Y.K.); w200r23@sch.ac.kr (Y.-B.C.); khlee@sch.ac.kr (K.-H.L.); dgmoon@sch.ac.kr (D.-G.M.)

**Keywords:** quantum dot light-emitting diode, solution process, MoO_3_ nanoparticles, hole injection layer, charge balance

## Abstract

This paper presents a study that aims to enhance the performance of quantum dot light-emitting didoes (QLEDs) by employing a solution-processed molybdenum oxide (MoO_x_) nanoparticle (NP) as a hole injection layer (HIL). The study investigates the impact of varying the concentrations of the MoO_x_ NP layer on device characteristics and delves into the underlying mechanisms that contribute to the observed enhancements. Experimental techniques such as an X-ray diffraction and field-emission transmission electron microscopy were employed to confirm the formation of MoO_x_ NPs during the synthesis process. Ultraviolet photoelectron spectroscopy was employed to analyze the electron structure of the QLEDs. Remarkable enhancements in device performance were achieved for the QLED by employing an 8 mg/mL concentration of MoO_x_ nanoparticles. This configuration attains a maximum luminance of 69,240.7 cd/cm^2^, a maximum current efficiency of 56.0 cd/A, and a maximum external quantum efficiency (EQE) of 13.2%. The obtained results signify notable progress in comparison to those for QLED without HIL, and studies that utilize the widely used poly(3,4-ethylenedioxythiophene):poly(styrene sulfonate) (PEDOT:PSS) HIL. They exhibit a remarkable enhancements of 59.5% and 26.4% in maximum current efficiency, respectively, as well as significant improvements of 42.7% and 20.0% in maximum EQE, respectively. This study opens up new possibilities for the selection of HIL and the fabrication of solution-processed QLEDs, contributing to the potential commercialization of these devices in the future.

## 1. Introduction

Since the groundbreaking report by Colin et al. in 1994 [1], quantum dot (QD) light-emitting diodes (QLEDs) have captured significant attention in the field of display and solid-state lighting applications. These devices have garnered interest due to their exceptional characteristics, including a narrow spectral emission bandwidth, size-tunable emission wavelength without altering the QD composition, high efficiency, and a low-cost fabrication technique compatible with solution-processed methods [2,3,4,5,6,7,8,9]. Over the years, rapid technological advancements have propelled the field of QLEDs forward. These advances mainly originated from the development of improved QD materials, inorganic charge transport materials, and more efficient device structures. Furthermore, a deeper understanding of the underlying device physics and manufacturing processes has played a crucial role in enhancing QLED performance and expanding their potential applications. [10,11,12,13,14,15]. Solution-processed devices offer compatibility with low-cost mass production and versatility, making them suitable for developing large-area flexible devices. These devices meet the demands of energy saving and diverse display requirements.

Several recent studies have reported various approaches to enhancing the performance of QDs themselves. These approaches include raising the valence band maximum (VBM) of QDs by applying to them a thin film of polyethylenimine ethoxylated to form a dipole, with the aim of adjusting the charge balance [16]. Another method involves improving stability and reducing non-radiative energy recombination by introducing an aluminum-doped shell layer of Al_2_O_3_ onto the surface of QDs [17]. The combination of multilayer CdSe@ZnS quantum dots (QDs) with 1,2-ethanedithiol solid-state treatment in the sequential formation of the emissive layer enables the adjustment of charge balance, ensuring stable light emission and increasing luminous efficiency [18]. Furthermore, the passivation of dangling bonds through the application of a chloride surface enhances mobility, resulting in high external quantum efficiency [19]. Additionally, efforts have been made to improve ligand quality, optimize thin-film characteristics, and refine charge balance to increase the quantum yield of photoluminescence, addressing the lower luminous efficiency of blue light-emitting QDs compared to red and green [20,21].

Achieving a balanced injection of holes and electrons is crucial in QLEDs due to the distinct transport behaviors of these charges. The injection of holes and electrons varies based on the difference in injection barrier height from the electrode to the device for each charge. Moreover, the transport behaviors of holes and electrons differ in terms of charge mobility and conductivity between the hole transport layer (HTL) and the electron transport layer (ETL). Extensive studies are being conducted on ETL and HTL to enhance the characteristics of QLEDs. In recent years, ZnO NPs have been widely used as ETLs in QLEDs. The presence of numerous oxygen vacancies in ZnO nanoparticle (NP) ETL can lead to nonradiative recombination at the interface with quantum dots (QDs). This nonradiative recombination can decrease device efficiency by reducing the number of excitons. To address this issue, a recent development involves the Mg-doping method, which helps reduce the number of oxygen vacancy states in ZnO [22].

In conventional QLED configurations, the energy barrier for hole injection from the anode to the QD through hole transport layer (HTL) is considerably greater than the energy barrier for electron injection from the cathode to the QD. Consequently, this leads to an imbalance of hole and electron carriers within the QD emitting layer (EML). To address this issue, a hole injection layer (HIL) can be introduced. The HIL serves to facilitate hole injection into the QD layer, thereby mitigating the charge imbalance and improving overall device performance.

The thin film of poly(3,4-ethylenedioxythiophene):poly(styrene sulfonate) (PEDOT:PSS) is the most widely employed organic HIL in QLEDs owing to its high conductivity, high work function, good thermal stability, high transparency, and electron-blocking ability. However, it is difficult to form a uniform interfacial contact between PEDOT:PSS and an indium tin oxide (ITO) electrode due to the hydrophilic nature of the PEDOT:PS. Because the PEDOT:PSS has hygroscopic and acidic properties, it can also lead to the corrosion of ITO electrode, affecting device performance by degrading device characteristics such as electroluminescence and lifetime [23,24,25]. In addition, due to the nature of organic materials, the PEDOT:PSS has a lower thermal stability than inorganic materials. Transition metal oxides such as molybdenum oxide (MoO_3_) [26,27,28], nickel oxide (NiO) [29,30,31], tungsten oxide (WO_3_) [32,33,34], and vanadium oxide (V_2_O_5_) [35,36,37] have successfully been employed in QLEDs as promising alternatives to replace the organic PEDOT:PSS HIL due to their compatibility with a high work function, good stability, and good carrier mobility. Consequently, transition metal oxides have applications in various devices, including photovoltaics, batteries, and light-emitting diodes. In particular, highly n-doped MoO_3_ has garnered significant attention as a promising material for HIL in QLEDs because it has a deep lying electronic state, efficient hole injection into organic material, and a wide bandgap [38,39,40,41,42]. Furthermore, the solution-processed MoO_3_ nanoparticles (NPs) exhibit good stability and compatibility with QD synthesis and device fabrication processes.

In this study, we focus on the integration of a solution-processed MoO_x_ NP HIL in QLED architecture to enhance its performance in terms of efficiency and stability. All layers except for the electrodes were fabricated using a solution-based process. We investigate the influence of varying the MoO_x_ NP concentration in the device characteristics. Moreover, we explored the underlying mechanisms responsible for the observed improvements in device performance, such as enhanced hole injection and improved charge balance. The structure and formation of MoO_x_ NPs during synthesis was confirmed using X-ray diffraction (XRD) and field emission–transmission electron microscopy (FE-TEM). The electronic structure of the QLEDs was analyzed using ultraviolet photoelectron spectroscopy (UPS). The QLED with MoO_x_ NPs at a concentration of 7 mg/mL achieved a maximum luminance of 69,240.7 cd/cm^2^, maximum current efficiency of 56.0 cd/A, and maximum external quantum efficiency (EQE) of 13.2%. These values represent a significant advancement compared to QLEDs without HIL and those utilizing the PEDOT:PSS HIL. They demonstrate a remarkable improvement of 59.5% and 26.4% in maximum current efficiency, respectively, and a significant enhancement of 42.7% and 20.0% in maximum external quantum efficiency (EQE), respectively. The findings present new possibilities for selecting hole injection layers and fabricating solution-processed QLEDs, paving the way for their future commercialization.

## 2. Materials and Methods

### 2.1. Synthesis of Materials

Figure 1 illustrates schematic illustrating the synthesis of MoO_x_ NPx. To synthesize MoO_x_ NPs, 1 g of molybdenum (Mo) powder (Sigma Aldrich, St. Louis, MO, USA) with a size of <150 nm and purity of 99.99% was added to a beaker that was placed in the ice bath, followed by slowly pouring 10 mL of hydrogen peroxide (H_2_O_2_) (Sigma Aldrich) solution into the beaker and stirring for 1 h. When the color of the solution changed from grey to orange, 10 mL of propionic acid (Sigma Aldrich) with a purity of 99.5% was added to the solution and stirred at 60 °C for 24 h to ensure that it was fully dissolved. MoO_x_ powder was obtained after vacuum distillation of the dissolved solution at 50 °C for 30 m. The powder was dispersed in ethanol (EtOH) (Daejung Chemical and Metals, Gyeonggi, Republic of Korea) at a pre-determined ratio.

We synthesized a colloidal suspension of zinc magnesium oxide Zn_0.9_Mg_0.1_O NPs using the sol–gel method [43].

### 2.2. Device Fabrication

The QLEDs were configured using MoOx NPs as the HIL, with the structure of ITO (150 nm)/MoOx NPs (x mg/mL: x = 0, 1, 3, 5, 7)/PVK (50 nm)/QD (10 nm)/ZnMgO (50 nm)/Al. Additionally, we fabricated a control device with an ITO/PEDOT:PSS/PVK (50 nm)/QD/ZnMgO/Al configuration, where PEDOT:PSS served as the HIL. QLEDs were fabricated using the following procedure. First, patterned indium tin oxide (ITO) thin films with a sheet resistance of ≤10 Ω/□ served as the anode. Prior to fabrication, the patterned ITO glass substrate underwent a sequential ultrasonic cleaning process using acetone, isopropyl alcohol, methanol, and deionized water. Subsequently, the patterned ITO glass substrate was spin-coated with four layers below:A solution of MoO_x_ NPs was spin-coated onto the ITO substrate at a speed of 2000 rpm for 20 s at room temperature.Poly(N-vinyl-carbazole) (PVK) (Sigma Aldrich) was dissolved at a concentration of 1.2 wt%. The PVK solution was then spin-coated onto the patterned ITO glass substrate. The spin-coating process consisted of spinning at 600 rpm for 5 s, followed by spinning at 1500 rpm for 15 s, both at room temperature. After the coating process, the substrate was annealed at 60 °C for 10 m.CdSe/ZnS QDs (Zeus) were dissolved in heptane at a concentration of 5 mg/mL to create the EML. The QD solution was spin-coated onto the ITO/PVK substrate at a speed of 3000 rpm for 5 s at room temperature.A solution of Zn_0.9_Mg_0.1_O NPs was spin-coated onto the ITO/MoO_x_/PVK/QD substrate at a speed of 2000 rpm for 20 s at room temperature.

After the spin-coating process, the multilayered substrates were loaded into a high-vacuum deposition chamber (Cetus OL 100; Celcose) with background pressure of 6 × 10^−7^ Torr). A 150 nm thick aluminum (Al) layer was deposited as the cathode using an evaporation rate of 1.2 Å/s. The Al cathode layer was patterned using an in situ shadow mask to form an active emitting area of 4 mm^2^.

### 2.3. Characterizations

The XRD patterns were measured using an X-ray diffractometer (D/Max-2200pc; Rigaku) in the 2θ range of 10° to 60° with a Cu-Kα (λ = 1.5405 Å) to confirm the formation of MoO_x_ NPs. FE-TEM images were obtained using a JEOL Tecnai F30 S-Twin microscopy UPS measurements were carried out using an X-ray photoelectron spectroscopy system (K-alpha; Thermo Fisher Scientific, Waltham, MA, USA) equipped with a He (I) 21.22 eV gas discharge lamp to analyze the O1s level and valence band maximum (VBM) of the MoO_x_ NP thin films. Transmittance and reflectance measurements were performed using a Shimadzu UV-1650PC spectrophotometer with monochromatic light incident on the sample surface. Current density–voltage–luminance (J-V-L) characteristics were evaluated using a computer-controlled source meter (Keithley Instruments, Cleveland, OH, USA) and a luminance meter (LS 100; Konica Minolta, Tokyo, Japan). Electroluminescence (EL) spectra were recorded using a Konica Minolta CS1000 spectroradiometer for spectral analysis.

## 3. Results and Discussion

Figure 2 presents the XRD patterns of the MoO_x_ NPs obtained by scanning the 2θ range from 10° to 60°. These XRD patterns were analyzed to investigate the crystal structure and crystallite size of MoO_x_ NPs. The XRD patterns of the MoO_x_ NPs exhibit an orthorhombic structure, which was confirmed by comparing them with the Joint Committee on Powder Diffraction Standards (JCPDS Card No. 5-0508). A slight shift in the diffraction position and intensity of the MoO_x_ NPs is observed, suggesting a small distortion in the lattice caused by oxygen vacancies, leading to a change in interatomic spacing. The average crystallite size of the MoO_x_ NPs was estimated to be 3.60 nm using the Debye–Scherrer equation [44].

Figure 3 displays a FE-TEM image of MoO_x_ NPs. The average size of the MoO_x_ NPs was determined to be approximately 3.57 nm, which closely correlates with the results obtained from the XRD analysis.

Figure 4a,b present the UPS spectra of the valence band edge and secondary electron cutoff regions for various materials, including ITO, PEDOT:PSS, MoO_x_ NPs, PVK, QD, and ZnMgO NPs, to investigate their electronic structures. The spectra were normalized to facilitate comparison. The work function (φ) was determined using the equation φ = hν-(E_cutoff_ − E_onset_), where hν represents the photon energy (21.22 eV) of the He source and E_Fermi_ denotes the Fermi level. The estimated work function for ITO, MoO_x_ NPs at concentrations of 1 mg/mL, 3 mg/mL, 5 mg/mL, 7 mg/mL, 9 mg/mL, PEDOT:PSS, PVK, QD, and Zn_0.9_Mg_0.1_O NPs were estimated to be 4.38 eV, 4.69 eV, 4.77 eV, 4.79 eV, 4.88 eV, 4.66 eV, 4.30 eV, 3.98 eV, and 3.54 eV, respectively. As the thickness of MoO_x_ NPs increased, the work function also increased. This is consistent with previously reported results [42,45]. The onset energy in the valence band region (E_onset_), obtained from the UPS analysis, represents the energy difference between the Fermi level and VBM. Using the work function and E_onset_, the VBM values of MoO_3_ at concentration of 1 mg/mL, 3 mg/mL, 5 mg/mL, and 7 mg/mL were estimated to be 7.79 eV, 7.81 eV, 7.79 eV, and 7.84 eV below the vacuum level, respectively.

The optical band gaps of MoO_x_ NPs, PVK, QD, ZnMgO, and PEDOT:PSS were derived using the Tauc equation of (αhν)^2^ = A(hν − Eg) [45], where α, h, ν, and A represent the absorption coefficient, Planck’s constant, radiation frequency, constant coefficient, respectively, and E_g_ represents the optical bandgap. The calculated optical bandgap for the MoO_x_ NPs at concentrations of 1 mg/mL, 3 mg/mL, 5 mg/mL, 7 mg/mL, 9 mg/mL was determined as 3.49 eV, and the optical band gaps of PEDOT:PSS, PVK, QD, and Zn_0.9_Mg_0.1_O NPs were determined as 1.92 eV, 3.51 eV, 2.22 eV, and 3.76 eV, respectively, by extrapolating the linear portion of the non-linear curve to the *x*-axis. Using the optical bandgap estimated from UV–Vis absorption spectra and the VBM values, the conduction band maximum (CBM) levels of the MoO_3_ NPs at concentrations of 1 mg/mL, 3 mg/mL, 5 mg/mL, 7 mg/mL, PEDOT:PSS, PVK, QD, and Zn_0.9_Mg_0.1_O NPs were calculated as 4.28 eV, 4.31 eV, 4.30 eV, 4.37 eV, 3.52 eV, 2.25 eV, 4.95 eV, and 4.02 eV below the vacuum level, respectively. Figure 4c illustrates the energy level diagram of the QLEDs with the MoO_3_ NPs at concentrations of 1 mg/mL, 3 mg/mL, 5 mg/mL, 7 mg/mL, PEDOT:PSS HILs, PVK HTL, QD EML, and Zn_0.9_Mg_0.1_O NP ETL in thermal equilibrium. In the energy level diagram, it can be observed that electron injection from Al to the QD layer occurs immediately. In the case of n-doped semiconductors, electron extraction from the HOMO level of the PVK occurs through the MoO_x_ NP conduction band, followed by injection into ITO.

Detailed interfacial energy level diagrams, considering the energy band bending of ITO/MoO _x_ NPs/PVK based on the energy levels, are presented in Figure 5, and the energy level unit is omitted for clarity. The energy band bending is created by dipoles formed due to work function difference between ITO and MoO_x_ NPs as well as MoO_x_ NPS and PVK [3,42,45,46]. The dipoles formed at the interfaces rearrange the energy level alignment between ITO and PVK, resulting in improved hole injection and transport. The relatively low work function of ITO, compared to MoO_x_ NPs, raises the energy level of MoO_x_ NPs. Similarly, the relatively high work function of MoO_x_ NPs, in comparison to PVK, lowers the energy levels of PVK, leading to hole accumulation and band bending at the interface. The primary focus here is the alteration in the hole injection barrier. The energy level differences between the HOMO levels of PVK and the CBM levels of MoO_x_ NPs in the bulk region were found to be 2.2 eV, 2.59 eV, 2.65 eV, and 2.64 eV for MoO_x_ NP concentrations of 1 mg/mL, 3 mg/mL, 5 mg/mL, and 7 mg/mL in thermal equilibrium, respectively. However, the reduced energy level differences between the HOMO levels of PVK and the CBM levels at the junction of MoO_x_ NPs and PVK, resulting from band bending were determined to be 1.77 eV, 1.63 eV, 1.67 eV, and 1.58 eV for MoO_x_ NP concentrations of 1 mg/mL, 3 mg/mL, 5 mg/mL, and 7 mg/mL, respectively. As a result, the formation of dipoles at the interface significantly reduced the energy barrier for electron injection from the HOMO levels of PVK to the CBM levels of MoO_x_ NPs at the junction of MoO_x_ NPs and PVK. When an electric field is applied, electrons at the HOMO level of PVK can easily transit to the CBM level of MoO_x_ at the junction of MoO_x_ NPs and PVK. This is facilitated by the reduced energy level difference between the two, allowing for efficient electron injection. It can be understood that the injection of holes from ITO to PVK occurs through electron extraction from the HOMO level of PVK, passing through the CBM level of MoO_x_ NPs, and finally into ITO. This mechanism replaces the conventional process of hole transit from ITO through the VBM level of MoO_x_ NPs into the HOMO level of PVK, which is consistent with previously reported results [3,46]. The measured results indicate that, as the concentration of MoO_x_ NPs increased, there was a trend of decreasing energy barrier at the junction of MoO_x_ NPs and PVK for electron injection from PVK to MoO_x_ NPs. With a decrease in the energy barrier, the current density of the device should increase. However, the measured results indicate a decrease in the current density. This can be attributed to an increase in resistance caused by the thicker MoO_x_ NP layer, which in turn, reduces the electric field. It appears that the electric field effect has a greater impact on the current density than the energy barrier effect. As indicated in [47,48,49], when a thin layer of MoO_x_ NPs (1 mg/mL or 3 mg/mL) is employed, holes are directly injected into the HOMO level of PVK by tunneling from ITO via the MoO_x_ NP layer.

Figure 6 illustrates the current density and luminance curves as a function of voltage for the QLEDs with varying concentrations of MoO_x_ NP and PEDOT:PSS HILs. The turn-on voltages extrapolated from the J-V curves for the QLEDs with concentrations of 1 mg/mL, 3 mg/mL, 5 mg/mL, and 7 mg/mL of MoO_x_ NP, as well as PEDOT:PSS HILs were determined to be 0.5 V, 1.0 V, 1.5 V, 1.5 V, and 1.5 V, respectively. At an applied voltage of 16 V, the current densities of the QLEDs with MoO_x_ NP HILs at concentrations of 0 mg/mL, 1 mg/mL, 3 mg/mL, 5 mg/mL, 7 mg/mL, and 9 mg/mL were estimated to be 285.2 mA/cm^2^, 384.0 mA/cm^2^, 339.7 mA/cm^2^, 202.5 mA/cm^2^, 173.7 mA/cm^2^, and 0.0 mA/cm^2^, respectively. With the increase in MoO_x_ NP concentration, the thickness also increased, resulting in a decrease in the electric field, subsequently causing a reduction in current density. Notably, among all devices, the QLEDs with PEDOT:PSS HIL exhibited the highest maximum current density. Figure 6 also reveals that no current flowed when a voltage of 16 V was applied to the QLED with a 9 mg/mL concentration of MoO_x_ NP HIL.

The turn-on voltages at 1 cd/m^2^ for the QLEDs at concentrations of 1 mg/mL, 3 mg/mL, 5 mg/mL, 7 mg/mL, and PEDO:PSS were also extrapolated to be 4.68, 4.52, 4.46 V, 4.51 V, and 4.52 V, respectively. The turn-on voltage at 1 cd/m^2^ for the QLEDs without HILs, such as MoO_x_ NPs and PEDOT:PSS, was estimated to be 6.58 V. This demonstrates a significant reduction in the turn-on voltage at 1 cd/m^2^ with the presence of HILs, namely MoO_x_ NPs and PEDOT:PSS. The QLEDs with 0 mg/mL, 1 mg/mL, 3 mg/mL, 5 mg/mL, 7 mg/mL, and 9 mg/mL concentrations of MoO_x_ NP HILs achieved maximum luminance values of 71,993.7 cd/m^2^, 109,013.4 cd/m^2^, 111,781.8 cd/m^2^, 63,334.8 cd/m^2^, 69,240.7 cd/m^2^, and 0.0 cd/m^2^, respectively. It is observed that with the increase in the concentration of MoO_x_ NP HILs increased from 0 mg/mL to 3 mg/mL, both the current density and maximum luminance increased, leading to a deterioration in the charge balance in QD EML. On the other hand, when the concentrations of the MoO_x_ NP HILs reached 5 mg/mL and 7 mg/mL, the maximum luminance of the QLEDs decreased. Notably, among all the devices, the QLED with PEDOT:PSS HIL exhibited the highest maximum luminance of 143,510.7 cd/cm^2^.

Figure 7 illustrates current efficiency and EQE curves as a function of luminance for the QLEDs with different concentrations of MoO_x_ NP and PEDOT:PSS HILs. The maximum current efficiencies of the QLEDs with 0 mg/mL, 1 mg/mL, 3 mg/mL, 5 mg/mL, and 7 mg/mL concentrations of MoO_x_ NP, and PEDOT:PSS HILs were estimated to be 35.1 cd/A, 45.5 cd/A, 50.4 cd/A, 53.9 cd/A, and 56.0 cd/A, respectively, as shown in Figure 7a. However, the current efficiency of the QLED with a concentration of 9 mg/mL of MoO_x_ NP HIL did not have any current flow, thus preventing the measurement of current efficiency. It was observed that as the concentration of the MoO_x_ NP HILs increased from 0 mg/mL to 7 mg/mL, the current efficiency improved. These results suggest that, despite the increase in current due to the increase in thicknesses of MoO_x_ NP HILs, it is believed that the relative luminance was higher due to the charge balance of electrons and holes in the EML. In contrast, the QLED with PEDOT:PSS HIL exhibited a maximum current efficiency of 44.3 cd/A. This indicates that, while the QLED with PEDOT:PSS HIL achieved higher luminance, it also exhibited higher current density compared to the QLEDs with MoO_x_ NP HILs, resulting in an inferior charge balance.

Figure 7b illustrates the current efficiency of the QLEDs with MoO_x_ NP HILs at various concentrations of 0 mg/mL, 1 mg/mL, 3 mg/mL, 5 mg/mL, and 7 mg/mL, which were calculated to be 9.3%, 10.3%, 12.5%, 12.9%, and 13.2%, respectively. As the concentration of the MoO_x_ NP HILs increased from 0 mg/mL to 7 mg/mL, the EQE also increased, indicating an enhanced charge balance in the QD EML. However, there was a decrease in luminance as the concentration increased from 3 mg/mL to 7 mg/mL. On the other hand, the QLEDs with PEDOT:PSS HIL exhibited a maximum EQE of 11.0%. Despite its higher luminance, the charge balance of the QLED with PEDOT:PSS is inferior to that of the QLEDs with 3 mg/mL, 5 mg/mL, 7 mg/mL MoO_x_ NP HILs. Table 1 summarizes the key parameters obtained from the QLEDs with MoO_x_ NP and PEDOT:PSS HILs.

Figure 8 illustrates the normalized photoluminescence (PL) spectrum of the CdSe/ZnS QD, along with EL spectra obtained from the QLEDs with various concentrations of MoO_3_ NP and PEDOT:PSS. Table 2 summarizes the characteristics parameters of these spectra. It is important to note that the EL peaks of the QLEDs, with both MoO_x_ NP and PEDOT:PSS HILs, are centered at 532 nm, exhibiting a blue-shift of 12 nm compared to the PL peak of CdSe/ZnS QD. The observed blue-shifts can be attributed to factors such as Föster energy transfer, dielectric dispersions, and the Stark effect occurring under high-voltage and -current conditions [50,51]. The EL spectra of the QLEDs exhibit no undesired parasitic features corresponding to PVK emission, ensuring a high color purity. This observation suggests that charge recombination predominantly occurs within the CdSe/ZnS QD EML. Furthermore, Figure 8 demonstrates that the full width at half maximum (FWHM) of the spectra decreases as the concentration of the MoO_x_ NP HIL increases. Notably, the FWHMs of the spectra for QLEDs with 5 mg/mL and 7 mg/mL MoO_3_ HILs are narrower compared to the QLED with PEDOT:PSS HIL. The absence of parasitic PVK emissions in the narrow spectra confirms that the device emissions are primarily caused by electron–hole recombination within the CdSe/ZnS QD EML. These findings can be attributed to a favorable charge balance between holes and electrons within the CdSe/ZnS QD EML.

## 4. Conclusions

We investigated the utilization of solution-processed MoO_x_ NPs as the HIL in QLEDs, aiming to enhance their efficiency and stability. The QLEDs incorporating solution-processed MoO_x_ NP as the HIL exhibited superior performance compared to those utilizing PEDOT:PSS. The XRD and FE-TEM analyses confirmed that MoO_x_ NPs with an average size of 3.57 nm were formed through the solution process. Based on the UPS and UV-Vis measurements, the energy level diagram at thermal equilibrium of the QLEDs was constructed. Efficient hole injection was confirmed via the process of electron extraction from the HOMO level of PVK HTL into the conduction band of MoO_x_ NP HIL, subsequently followed by injection into the ITO anode. Remarkably, the QLED with 8 mg/mL concentration of MoO_x_ NPs achieved outstanding performance metrics, including a maximum luminance of 69,240.7 cd/m^2^, a maximum current efficiency of 56.0 cd/A, and a maximum EQE of 13.2%.

The results represent a significant advancement compared to QLED without HIL and those using the widely used poly(3,4-ethylenedioxythiophene):poly(styrene sulfonate) (PEDOT:PSS) HIL. They demonstrate remarkable improvements of 59.5% and 26.4% in maximum current efficiency, respectively, as well as a significant enhancement of 42.7% and 20.0% in maximum EQE, respectively. Moreover, the simplicity and low-temperature compatibility of the solution-process preparation method for MoO_x_ NP thin films, employed in this study, make it suitable for flexible substrate applications. These findings demonstrate the promising potential of MoO_x_ NPs as the HIL materials, offering a practical and cost-effective platform for the advancement of large-area QLEDs with exceptional performance and stability.

## Figures and Tables

**Figure 1 nanomaterials-13-02324-f001:**
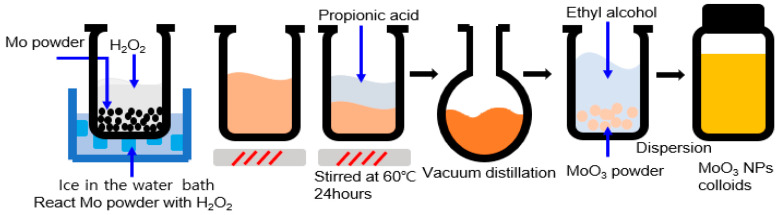
Schematic illustrating the synthesis process of MoO_3_ nanoparticles (NPs).

**Figure 2 nanomaterials-13-02324-f002:**
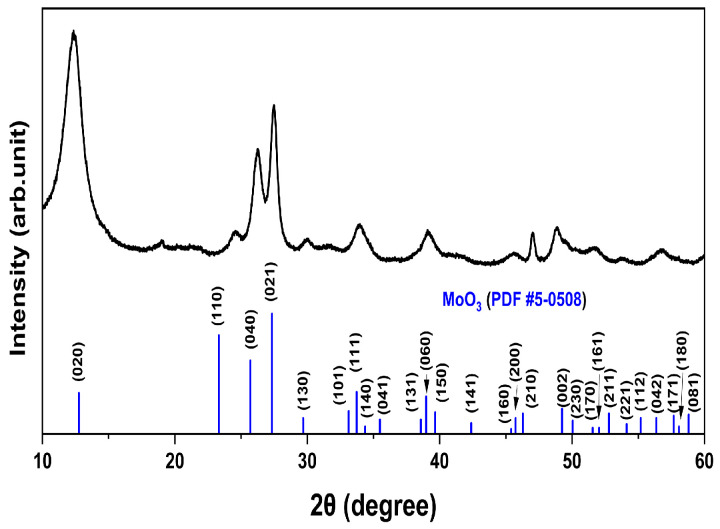
X-ray diffraction (XRD) patterns of the MoO_x_ NPs obtained by scanning the 2θ range from 10° to 60°.

**Figure 3 nanomaterials-13-02324-f003:**
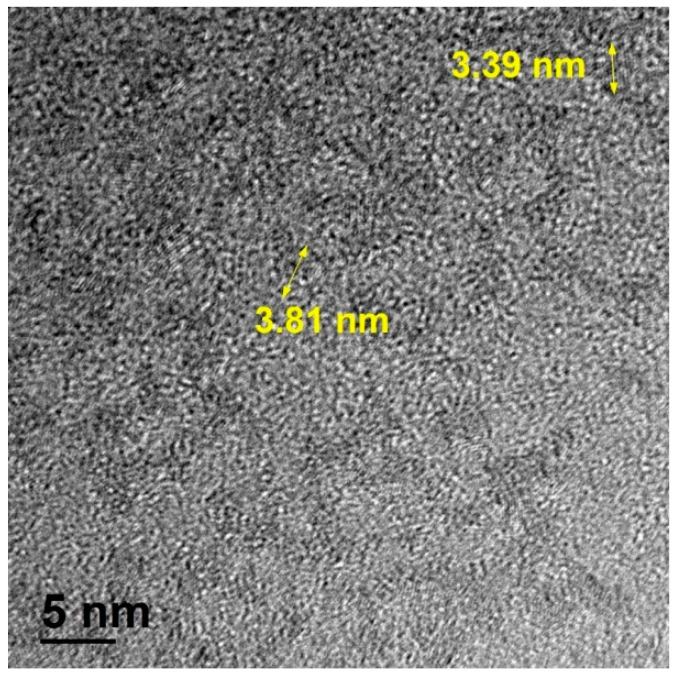
Field emission–transmission emission microscopy (FE-TEM) image of MoO_x_ NPs.

**Figure 4 nanomaterials-13-02324-f004:**
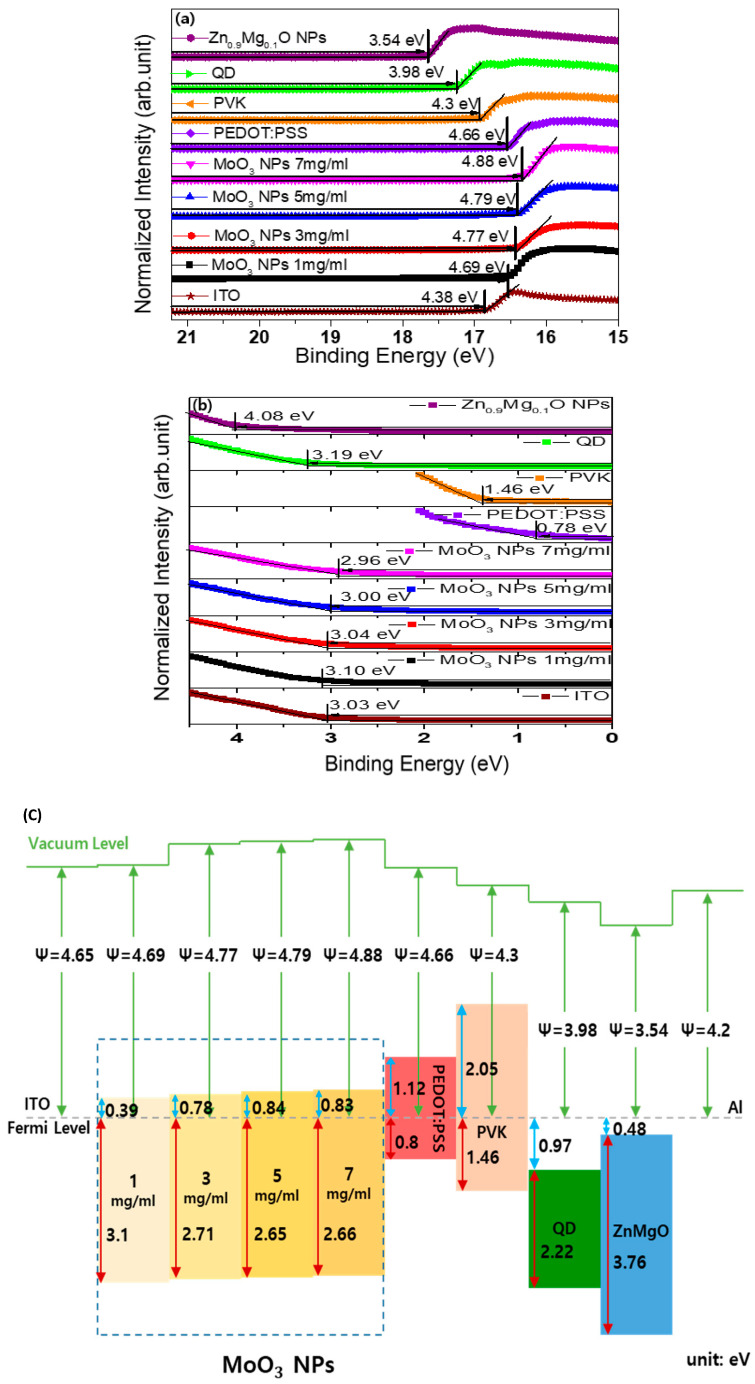
Ultraviolet photoelectron spectroscopy (UPS) spectra of (**a**) the valence band edge. (**b**) Secondary electron cutoff regions for various materials, including ITO, MoO_x_ NPs with different concentration, PEDOT:PSS, PVK, QD, and ZnMgO NPs. (**c**) Schematic energy level diagram of the QLEDs at thermal equilibrium.

**Figure 5 nanomaterials-13-02324-f005:**
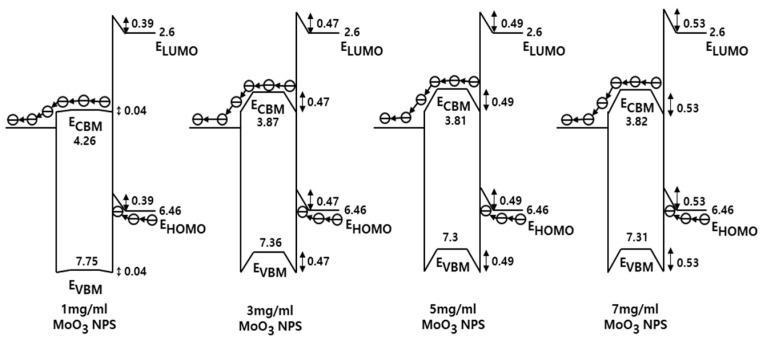
Energy band diagrams of ITO/MoO_x_ NPs (x = 1 mg/mL, 3 mg/mL, 5 mg/mL, 7 mg/mL)/PVK structures.

**Figure 6 nanomaterials-13-02324-f006:**
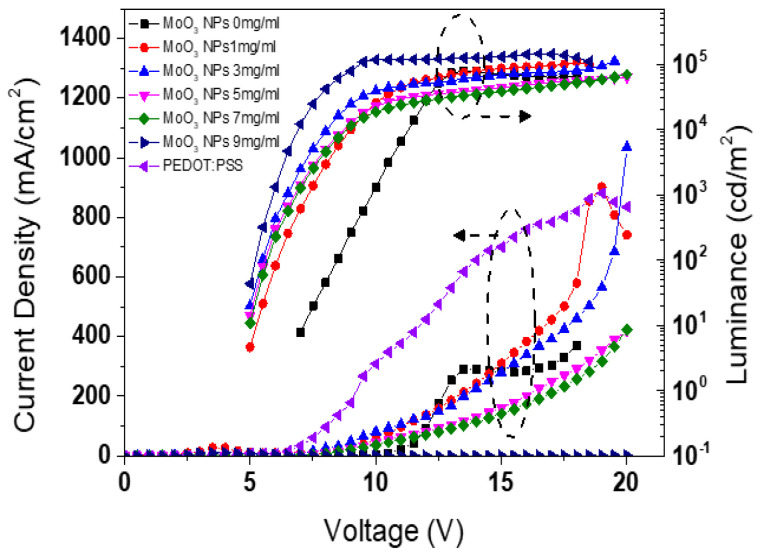
Current density and luminance curves as a function of voltage for the QLEDs with various concentrations of MoO_3_ NP and PEDOT:PSS HILs.

**Figure 7 nanomaterials-13-02324-f007:**
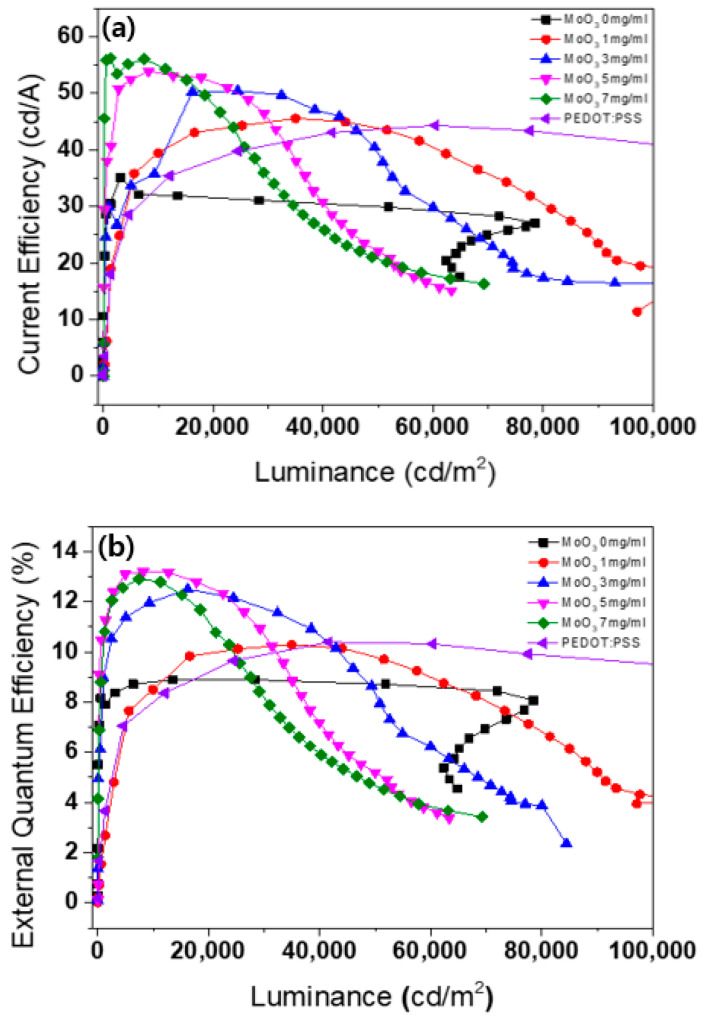
(**a**,**b**) Current efficiency and EQE curves as a function of the current density of the QLEDs with various concentrations of MoO_x_ NP and PEDOT:PSS HILs.

**Figure 8 nanomaterials-13-02324-f008:**
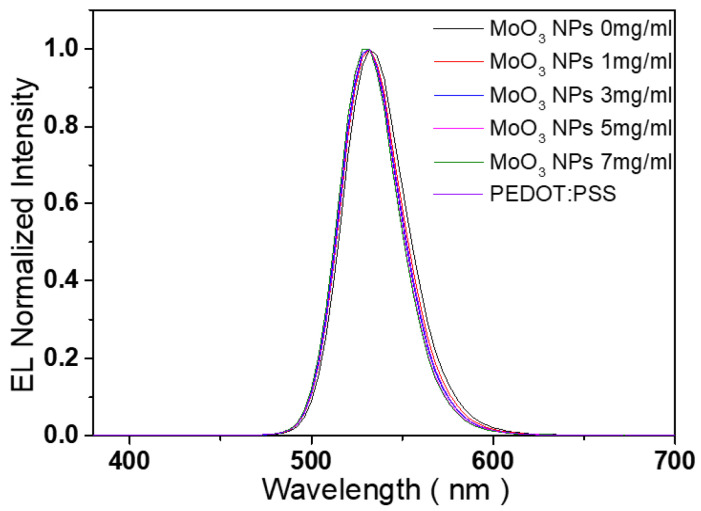
Normalized photoluminescence (PL) spectrum of the CdSe/ZnS QD and electroluminescence (EL) spectra of the QLEDs with various concentrations of MoO_x_ NPs and PEDOT:PSS.

**Table 1 nanomaterials-13-02324-t001:** The performance parameters of QLEDs with various concentrations of MoO_x_ NP and PEDOT:PSS hole injection layers (HILs).

Types of HIL	Current Density at 16 V	Maximum Luminance	Maximum Current Efficiency	Maximum EQE
	(mA/cm^2^)	(cd/m^2^)	(cd/A)	(%)
PEDOT:PSS	761.2	143,510.70	44.3	11
MoO_x_ NPs of 0 mg/mL	285.2	71,993.70	35.1	9.25
MoO_x_ NPs of 1 mg/mL	384	109,013.40	45.5	10.3
MoO_x_ NPs of 3 mg/mL	339.7	111,781.80	50.4	12.5
MoO_x_ NPs of 5 mg/mL	202.5	63,334.80	53.9	12.9
MoO_x_ NPs of 7 mg/mL	173.7	69,240.70	56	13.2

**Table 2 nanomaterials-13-02324-t002:** The key parameters of the spectra, including the photoluminescence of the QD and the electroluminescence of QLEDs with various concentrations of MoO_x_ NP and PEDOT:PSS HILs.

Sample	Peak (nm)	Fwhm (nm)
PL of QD	544	33.3
EL of QLED with 0 mg/mL MoO_x_ NP HIL	532	41.0
EL of QLED with 1 mg/mL MoO_x_ NP HIL	532	35.9
EL of QLED with 3 mg/mL MoO_x_ NP HIL	532	36.0
EL of QLED with 5 mg/mL MoO_x_ NP HIL	532	36.6
EL of QLED with 7 mg/mL MoO_x_ NP HIL	532	37.2
EL of QLED with PEDOT:PSS HIL	532	39.6

## Data Availability

The data that support the findings of this study are available from the corresponding authors upon reasonable request.

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
