# Peer review of "Highly Efficient All-Solution-Processed Quantum Dot Light-Emitting Diodes Using MoOx Nanoparticle Hole Injection Layer"

_nanomaterials, 2023, doi:10.3390/nano13162324_

Round 1

Reviewer 1 Report

The paper “Highly Efficient All-Solution Processed Quantum Dot Light-Emitting Diodes Using MoO3 Nanoparticle Hole Injection Layer” presents a solution-processed molybdenum oxide (MoO3) nanoparticle as a hole injection layer enabling improvements in device performance. This work is sufficiently thorough and would interesting for the readers of Nanomaterials; however, there are deficiencies (listed below) that should be addressed before acceptance for publication. My detailed comments are as follows:

Q1: The author claimed they synthesized the MoO3 nanoparticles, in fact the XRD of which is not a good match (Figure 2), the author should determine whether HIL is MoO3 or MoOx.

 Q2: Changes in MoO3 concentration affect the thickness of MoO3 layer in QLED. Why do UPS results show changes in work function and optical bandgap? This is very confusing.

 Q3: This manuscript shows that the QLEDs with different concentrations of MoO3 influence the EL peak width. The reason needs to be given.

 Q4: As the concentration of the MoO3 NP HILs increased from 0 mg/mL to 7 mg/mL, the corresponding EQE values have also been increased, indicating an enhanced charge balance in the QD EML. Why does the device luminance decrease so much compared to the device based on MoO3 with concentration of 3 mg/mL (Table 1)?

 Q5: In general, the ratio of CE to EQE for QLED is similar when the EL peak position or FWHM have not been changed. However, in this manuscript, the QLEDs show different CE/EQE ratio with 3 mg/mL and 5 mg/mL MoO3 as the EL peak position and FWHM of these two devices show very similar values.

 Q6: It is suggested that the authors add some recent progress descriptions about inorganic HTL based QLEDs (such as Adv. Mater. 2022, 34, 2108150; Small 2021, 17, 2100030; Adv. Photonics Res. 2021, 2, 2000124) and highly efficient Cd-free QLEDs (Advanced Science, 2022, 9, 2200959) in the introduction to increase the readability of the manuscript.

The English writing should be further improved.

Author Response

Response to Reviewer 1

The authors are grateful for the valuable comments and suggestions from the reviewer about our manuscript (nanomaterials-2465879). We have addressed the comments raised by the review, and the amendments are highlighted in yellow in the revised manuscript. Our detailed responses to the reviewers’ comments are as follows.

Reviewer 1:

The paper “Highly Efficient All-Solution Processed Quantum Dot Light-Emitting Diodes Using MoO3 Nanoparticle Hole Injection Layer” presents a solution-processed molybdenum oxide (MoO3) nanoparticle as a hole injection layer enabling improvements in device performance. This work is sufficiently thorough and would interesting for the readers of Nanomaterials; however, there are deficiencies (listed below) that should be addressed before acceptance for publication. My detailed comments are as follows:

Question 1) The author claimed they synthesized the MoO3 nanoparticles, in fact the XRD of which is not a good match (Figure 2), the author should determine whether HIL is MoO3 or MoOx.

Response 1) We are very grateful for your good indication. The position of the peak does not exactly match the given peak in the JCPDS. Therefore, it is more accurate to refer to it as MoOx. We have revised MoO3 to MoOx in this manuscript.

Question 2: Changes in MoO3 concentration affect the thickness of MoO3 layer in QLED. Why do UPS results show changes in work function and optical bandgap? This is very confusing.

Response 2) We thank the reviewer’s valuable advice. There were several papers indicating that the work function can be changed due to the formation of a dipole at the interfaces when the thickness of MoOx varies [1-3]. We have revised the manuscript by adding the references of [1-3]. Considering the very small changes in the optical bandgaps of MoOx NPs that we measured, which fall within the range of experimental error and are negligible, we have revised the manuscript following your suggestion. We used the average value of 3.49 eV to ensure consistency.

[1] M. Kröger, S. Hamwi, J. Meyer, T. Riedl, W. Kowalsky, and A. Kahn, Role of the deep-lying electronic states of MoO3 in the enhancement of hole-injection in organic thin films, Appl. Phys. Lett. 95, 123301 (2009).

[2] Irfan, H. Ding, Y. Gao, D. Y. Kim, J. Subbiah, F. So, Energy level evolution of molybdenum trioxide interlayer between indium tin oxide and organic semiconductor, Appl. Phys. Lett. 96, 073304 (2010).

[3] K. Kanai, Kenji Koizumi, S. Ouchi, Y. Tsukamoto, K. Sakanoue, Y. Ouchi, K. Seki, Electronic structure of anode interface with molybdenum oxide buffer layer, Org. Electron. 11 (2010) 188-194.

Question 3: This manuscript shows that the QLEDs with different concentrations of MoO3 influence the EL peak width. The reason needs to be given.

Response 3) We thank reviewer’s good advice. We are sorry that there were typographical errors in the manuscript. We revised the errors. It was observed that as the thickness of MoOx increases, the FWHM decreases very slightly. We think that this phenomenon was attributed to the improvement in charge balance in the QD EML as the thickness increases.

Question 4: As the concentration of the MoO3 NP HILs increased from 0 mg/mL to 7 mg/mL, the corresponding EQE values have also been increased, indicating an enhanced charge balance in the QD EML. Why does the device luminance decrease so much compared to the device based on MoO3 with concentration of 3 mg/mL (Table 1)?

Response 4) We thank you for your good indication. When the concentration of MoOx increases from 1 mg/mL to 7 mg/mL, the current density decreases from 384.0 mA/cm2 to 173.7 mA/cm2. To achieve higher luminance, more exciton formation due to electron-hole recombination is required. For this purpose, a larger current is necessary. Therefore, as the current density has decreased, the luminance decrease as well. We think that the current efficiency and EQE values increased because the rate of luminance decrease is smaller than the rate of current decrease when the MoOx concentration increases from 1 mg/mL to 7 mg/mL.

Question 5: In general, the ratio of CE to EQE for QLED is similar when the EL peak position or FWHM have not been changed. However, in this manuscript, the QLEDs show different CE/EQE ratio with 3 mg/mL and 5 mg/mL MoO3 as the EL peak position and FWHM of these two devices show very similar values.

Response 5) Thank you for good indication. According to revised data, we consider the CE/EQE ratio using QLEDs with 3 mg/mL and 5 mg/mL MoOx HILs to be similar, with values of 4.04 and 4.17, respectively.

Question 6: It is suggested that the authors add some recent progress descriptions about inorganic HTL based QLEDs (such as Adv. Mater. 2022, 34, 2108150; Small 2021, 17, 2100030; Adv. Photonics Res. 2021, 2, 2000124) and highly efficient Cd-free QLEDs (Advanced Science, 2022, 9, 2200959) in the introduction to increase the readability of the manuscript.

Response 6) Thank you for your valuable suggestion. Following your advice, we have included the references in this manuscript to reflect state-of-the-art trend.

Reviewer 2 Report

Quantum dot based LEDs are, without doubt, a focus of a strong research in last years. Combining the device structure simplicity with a well-known high emission, can be clearly use in everyday more wide practical applications. In such line, the several current work made, points to very interesting devices figures of merit. However, some results needs to be clarified, explained and framed in suitable physical models, instead only in a collection of data, that will not impact in future works.

The current manuscript, shows some interesting results of solution deposited QD-LEDs based on CdSe/ZnS QDs. The figures of merit are quite impressive, although the anode optimization with an injection layer modulated with MoO3 is not surprising and not new. In spite of the presented data, it is my opinion that the manuscript has a major flaw in interpretation, exploration of physical models and discussion of results, which prevent a clear use of the results. Moreover, there are some absence of data that can be useful for a better understand of the manuscript. In this line of thought, I understand that the manuscript, in its current form, is not suitable for publication in Nanomaterials without a thorough review. In this review, authors should take into account the following aspects:

a) The authors needs to (re)write completely the introduction, making a real state of the art concerning the QD-LEDs based on CdSe/ZnS, even built by solution deposition process;

b) Same for use of MoO3 HIL. There are innumerous published works about the effect of modulation of MoO3, in particular regarding the thickness;

c) In the description of the ZnMg NPs synthesis, the authors needs to explain the values (ratios, mass and solution concentrations) used. Ref 39 does not explain and, by the way, the authors needs to discuss the real improvements of the current work by comparison with the published in ref 39;

d) The authors needs to indicate all the deposition procedure, including how the solvent as removed from all layers (and indicate what was the solvent for some layers). Also, the thickness of each layer needs to be made (such data will impact in the electrical balance equilibrium of the device). Without that, there makes no sense discuss any data based on thickness;

e) A detailed explanation need to de made about how the EQE was calculated; moreover, how was (eventually) the area correction when using the Minolta Luminance Meter to obtain (extrapolate!) the bright;

f) It is well-known the eventual environment issues for a correct UPS data measurement. Did the authors take any particular care? Also, in the figure 4 (a, b and c) how the data about the MoO3/9 ml NPs is not presented as well the work-function?

g) The work function value for ITO is extremely low compared to what the literature reports. It should be revised considering the usual difficulties in the UPS technique;

h) The explanations given for the charge injection process are particularly contradictory with the energy levels scheme in Figure 4. Even considering that the ITO work function may be the right one, PEDOT:PSS presents a much lower barrier to the holes injection than any of the MoO3 layers. This whole part of the manuscript has to be reviewed, explained and properly discussed based on the energy diagram (but not as it is).

i) The turn-on voltage is obviously the applied potential for which the device starts to emit (usually 1cd/m2);

j) In figure 5, from the J-V data, it seems to me that there are interlayers that don't suffer from "resistive switching". This usually indicates space charge issues. The authors needs to study in detail the SCLC behavior of their devices, in order to estimate the carrier’s electrical mobility. In such approach, a simple study of the defects acting as traps for electrical carriers should be an interesting point to discuss how effective is the electrical balance efficiency. With the energy levels shown, is not easy;

k) A plot of EQE (and current efficiency) versus bright will be important to estimate the roll-off (that in turn can give important information about the electrical balance efficiency. The high value of EQE is only achieved immediately after the LED turn-on;

l) The EL spectrum should be made for different applied voltages, in order to observe the electrical carrier confinement inside the active layer;

The authors should consider minor grammatical corrections and eventually rephrase some sentences that appear to be somewhat complex, in order to better explain the ideas and concepts.

Author Response

Response to Reviewer 2

The authors are grateful for the valuable comments and suggestions from the reviewer about our manuscript (nanomaterials-2465879). We have addressed the comments raised by the review, and the amendments are highlighted in yellow in the revised manuscript. Our detailed responses to the reviewers’ comments are in attached file. 
